# Design and degradation of permanently porous vitamin C and zinc-based metal-organic framework

Tia K. Tajnšek[1,2], Erik Svensson Grape[3], Tom Willhammar [3], Tatjana Antonić Jelić[4], Uroš Javornik[1], Goran Dražić [1], Nataša Zabukovec Logar [1,5] & Matjaž Mazaj [1✉]

Bioapplication is an emerging field of metal-organic frameworks (MOF) utilization, but bio-compatible MOFs with permanent porosity are still a rarity in the field. In addition, bio-compatibility of MOF constituents is often overlooked when designing bioMOF systems, intended for drug delivery. Herein, we present the a Zn(II) bioMOF based on vitamin C as an independent ligand (bioNICS-1) forming a three-dimensional chiral framework with permanent microporosity. Comprehensive study of structure stability in biorelevant media in static and dynamic conditions demonstrates relatively high structure resistivity, retaining a high degree of its parent specific surface area. Robustness of the 3D framework enables a slow degradation process, resulting in controllable release of bioactive components, as confirmed by kinetic studies. BioNICS-1 can thus be considered as a suitable candidate for the design of a small drug molecule delivery system, which was demonstrated by successful loading and release of urea—a model drug for topical application—within and from the MOF pores.

[1] National Institute of Chemistry, Hajdrihova 19, 1000 Ljubljana, Slovenia. [2] Faculty of Inorganic Chemistry and Technology, University of Ljubljana, Večna pot 113, 1000 Ljubljana, Slovenia. [3] Stockholm University, Frescativägen 8, 106 91 Stockholm, Sweden. [4] Ruđer Bošković Institute, Bijenička cesta 54, 1000 Zagreb, Croatia. [5] University of Nova Gorica, Vipavska 13, 5000 Nova Gorica, Slovenia. ✉email: matjaz.mazaj@ki.si

Metal-organic frameworks (MOFs) are crystalline and generally porous materials consisting of metal ions coordinated to organic molecules, capable of forming 2D and 3D structures with specific structural features. Large pore volume, high surface-to-volume ratio, the possibility of pore functionalization and the use of active constituents, are properties commonly attributed to these materials. MOFs are already well established in the various fields of science, such as catalysis, gas storage and separation, sensing or electrochemistry and they are increasingly gaining interest in the biomedicine applications as well[1–5]. The compositional variety of these materials allows us to create systems with an agreeable toxicological profile for the use of drug delivery, contrast agents and theranostics[6–10]. Such MOFs are, if constructed from at least one biologically active unit, further on referred to as bioMOFs[11–13].

From the perspective of MOF design, Zn(II) is probably one of the most frequently used cations. Its electronic configuration namely enables a variety of coordination geometries ranging from four- to sixfold polyhedra and thus the formation of many different secondary building units (SBUs), most commonly found either as discrete clusters or infinite rod-like shapes[14,15]. However, 3D infinite SBUs forming inorganic skeletons, which have been observed for Mn-, Ni- or Mg-based MOFs for instance, are to the best of our knowledge not yet known for the Zn-based frameworks[16–18]. From the biological perspective, Zn(II) is an essential element with recommended daily intake (RDI) value of ~10 mg and a crucial component of a large number of enzymes and transcriptional factors[19,20]. It has a significant role in response to oxidative stress, DNA replication and damage repair, as well as immune response, apoptosis, aging and synthesis of proteins, for example, collagen[21]. Zinc's toxicity has also been well studied and oral lethal dose (represented by $LD_{50}$ value) was determined to be close to 3 g kg$^{-1}$ of body weight in humans[22,23]. In contrast to Fe(III) being most frequently used in the design of bioMOFs, Zn with lower RDI than Fe does not have the tendency to bio-accumulate in the body caused by the lack of active physiological excretion pathways[21,24]. Moreover, Zn is regarded as an antioxidant mineral, is redox inactive and as a cation (within RDI values) does not promote the generation of undesirable reactive oxygen species (ROS)[19,21]. Its antimicrobial activity is depended on concentration and time of contact. Zinc's effect is a consequence of (i) direct interaction with the microbial membrane leading to its destabilization and (ii) interaction with nucleic acids and deactivation of the respiratory system[25]. Therefore, Zn can present a suitable candidate in constructing bioMOF systems. When designing a bioMOF by using a desired drug or active molecule as a linker, achieving permanent porosity, although desirable, is not always a priority. This is well exampled in a nonporous bioMIL-5 where both constituents, azelaic acid and Zn(II) are used for the antibacterial treatment of skin disorders and are released in a controllable manner[26]. Permanent porosity, with the pores accessible for hosting molecules after solvent removal, would however offer additional bioapplicative opportunities. The lack of permanent porosity or inaccessibility of the pores for drug molecules in bioMOF structures, when using small endogenous molecules as ligands, is considered to be the main drawback in their development[27]. There are however a few examples of porous bioMOFs with therapeutic linkers worth mentioning—(i) analogue Mg-MOF-74, where olsalazine (prodrug of the anti-inflammatory 5-aminosalicylic acid)[28] replaced 2,5-dihydroxybenzene-1,4-dicarboxylate (DBDC) as a linker; (ii) medi-MOF-1[29], which utilizes curcumin (anti-oxidative, anti-inflammatory, anticancer activity) and (iii) bioMIL-3[30], which employs 3′,5′5-azobenzenetetracarboxylate (antimicrobial) as a linker, however, this framework selectively adsorbs $CO_2$ and NO and is not otherwise considered porous.

Besides pharmaceutically active ingredients, there are different groups of organic bio-molecules with abilities to coordinatively bind with metal cations or SBUs, such as amino acids, nucleobasis, porphyrins or sugars[31–38]. Vitamins with natural abundance and a variety of biomedical applications have the ability to bind metals through multiple coordination sites as well. In spite of the apparent advantages that this group of biomolecules offers, vitamin B$_3$ (niacin/nicotinic acid) is, to the best of our knowledge, the only example used as a ligand in designing bioMOFs[39–41]. In that manner, vitamin C (L-ascorbic acid—ASC) also presents a suitable, biologically active linker. Just as vitamin B$_3$, ASC is also water-soluble and has anti-inflammatory and antioxidant properties however, their pharmacology is very different. Vitamin C is present in plasma and tissues, and presents an essential cofactor in numerous enzymatic reactions (biosynthesis of collagen, carnitine, and neuropeptides)[42]. Together with other vitamins and trace elements such as Zn improves immune functions[21]. Vitamin C is a weak sugar acid (pKa1 = 4.25 and pKa2 = 11.79) that forms a monoanion (HA) at pH 4–5 and dianion (A) at pH 11–12. Various materials where metal cations are combined with ascorbic acid are known, yet they possess molecular or 1D chain-like structures or use ascorbic acid only as a sacrificial agent for sensing applications[43–51]. Therefore, a three-dimensional bio-MOF framework composed of ascorbic acid as an independent ligand was not known up until now.

Herein we present Zn-based ascorbate MOF (bioNICS-1; bio-MOF material of National Institute of Chemistry, Slovenia), possessing a unique structure with 3D infinite inorganic building unit inter-connected through ascorbate ligands. BioNICS-1 presents, to the best of our knowledge, the first example of ascorbate with permanent microporosity, enabling the encapsulation of small drug molecules. Insight into the degradation of Zn-ascorbate structure in biorelavant media shows great potential for controlled drug release.

## Results and discussion

BioNICS-1 crystallizes from solvothermal conditions using only ethanol as a solvent. Ethanol was chosen for the synthesis as it is technologically safe with a relatively low impact on human health and the environment, and is so considered as a green solvent[52–54]. Furthermore, ethanol acts as a weak ligand and thus has low tendency to coordinatively participate in the formation of framework structure. Consequently, its removal to activate the MOF framework is typically easier than in the case of the use of stronger-ligand solvents such as water or dimethylformamide.

**Structure properties**. A broader size scale of crystals can be achieved either by using modulator (acetic acid) resulting in the formation of individual crystallites with the size up to 2 μm (bioNICS-1-aa), or the choice of the heating source (microwave) for the synthesis of nanoparticles (bioNICS-1-mw) with dimensions up to 50 nm. The details of the synthesis procedures are described in the "Methods" section. The particle size of differently prepared bioNICS-1 products was comparatively estimated by SEM (Supplementary Fig. 1) and Scherrer calculations (Supplementary Table 1). The results of dynamic light scattering (DLS) measurements were not satisfactory for proper particle size distribution analysis due to the agglomeration tendencies of the nanocrystals[55], which can be explained by the low surface charge deducted by Zeta potential measurements (Supplementary Fig. 2). Upon stirring the suspension of bioNICS-1 in water media, sedimented material does re-disperse—whether or not these are original agglomerates or newly formed ones remains unclear. Agglomeration of nanoparticles in liquid media is a common occurrence and could be later addressed with surface modifications

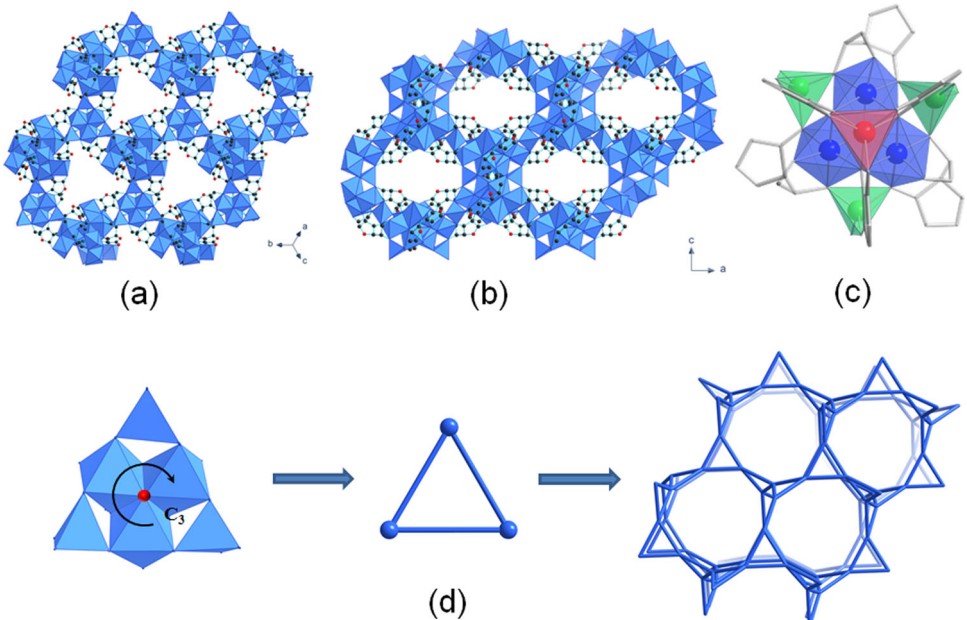

**Fig. 1 Schematic representation of bioNICS-1 structure.** Array of inorganic building unit (shown as blue polyhedra) forms two types of channels which are viewed **a** along [111] and **b** along [010] directions and are ascribed in the manuscript as type B and A respectively. Ascorbic acid linker molecules are represented with C-atoms as black dots and O-atoms as red dots. Hydrogen atoms are omitted for clarity. **c** Fragment of the structure building unit representing three types of Zn(II) coordination environment—octahedral (blue polyhedra), tetrahedra (green polyhedra) and trigonal prismatic (red polyheder). Ascorbates linking the inorganic building unit are shown in grey skeletal mode. **d** Topologyical interpretation of the framework. Triangular inorganic building unit motif shown in left-most figure possess centre of C3 symmetry represented by O atom belonging to OH group in Zn(ASC)(OH) structure (red circle). This motif can be simplified as triangles (middle figure) constructing the framework with lcv topology (right figure).

to provide either greater surface charge or steric interference[56]. The crystal structure of bioNICS-1 was determined by using 3D electron diffraction (3DED) (Supplementary Fig. 3, Supplementary Table 2) and further refined with PXRD (Supplementary Fig. 4, Supplementary Table 3). For electron diffraction study bioNICS-1-aa was used, whereas all the remaining investigations were performed on bioNICS-1 product synthesized under conventional conditions followed by the activation described in the Supplementary methodes. Results of PXRD, TGA and $N_2$ isotherms deemed these two materials to be the same (Supplementary Fig. 5). The crystal structure contains infinite three-dimensional inorganic building unit bridged through the ascorbate moieties, with Zn(II) occurring in three different coordination environments i.e. octahedral, trigonal prismatic and tetrahedral (Fig. 1c). Such coordination diversity of Zn(II) cation within one framework structure is a rarity among MOFs and is a consequence of the multi-dentate property of ascorbic acid being fully deprotonated with all four hydroxyl and additional carbonyl O atom participating in coordination with Zn(II) centres (Supplementary Fig. 6). An additional O atom that shares a common vertex of three $ZnO_6$ octahedra is subjected to C3 symmetry (Fig. 1d) and belongs to the independent hydroxyl group, yielding the overall chemical formula of $Zn_3(ASC)$ (OH). The chirality of ascorbic acid results in a chiral framework, as bioNICS-1 crystallizes in the space group $I2_13$. The arrangement of Zn(II)-centred polyhedra can be simplified as triangular motifs with tetrahedra on their corners, resulting **lcv** topology (Fig. 1d)[57]. Even though the ascorbate ligands are not part of the topological description, their role in linking of the cations is crucial for the formation of the framework, since the absence of ascorbate does not yield any precipitate nor is this structure possible to obtain with a different linker. A similar arrangement of inorganic units can be found for example in MIL-77(Ni) structure[17], but for Zn-based SBUs this is unprecedented. A complex system of polyhedral inorganic chains forms two types of intersecting channels with type

A running along each unit cell axis and type B running along [111] defined by eight-membered and six-membered rings, having diameters of 6.5 and 5.5 Å respectively (Fig. 1a, b).

The open framework structure of bioNICS-1 was confirmed by $N_2$ sorption isothermal measurements. Type I isotherm classified by IUPAC indicates permanent microporosity with BET surface area of 553 $m^2\,g^{-1}$ (Fig. 2a, Supplementary Figs. 7 and 8) and micropore volume of 0.22 $cm^3\,g^{-1}$. Pore size distribution using NLDFT method shows a bimodal distribution of micropores with the peaks centred at 5.6 and 6.6 Å belonging to B and A channel types respectively, already described above. BioNICS-1 so represents the first case of vitamin C-based MOF material with proven permanent microporosity. Moreover, the high density of Zn(II) sites within bioNICS-1 structure suggests the acid nature of the MOF framework. Indeed, results of temperature-programmed desorption (TPD) of ammonia shown in Fig. 2b and Supplementary Fig. 9 confirms that approximately 1/3 of the total Zn(II) cations (2.6 mmol $g^{-1}$) that builds the structure represent mostly weak acid sites that are potentially accessible for therapeutic gasotransmitters, such as nitric oxide (NO), hydrogen sulfide ($H_2S$) or carbon monoxide (CO)[58].

**Structure solubility in aqueous media.** Due to the open structure, bioNICS-1 is considered as a potential drug carrier. In addition, because of the nature of building units, the framework itself can represent a source of bioactive components, which can have, with suitable dosing, a therapeutic effect.

The material was first tested for structure thermal stability (Supplementary Figs. 11 and 12) and solubility in water media (see Supplementary Note 1 for details) in order to investigate the release of the material's constituents. To approximate bioNICS-1 behaviour to realistic conditions of human organism-environment, the tests included the influence of three different parameters: (a) the possible

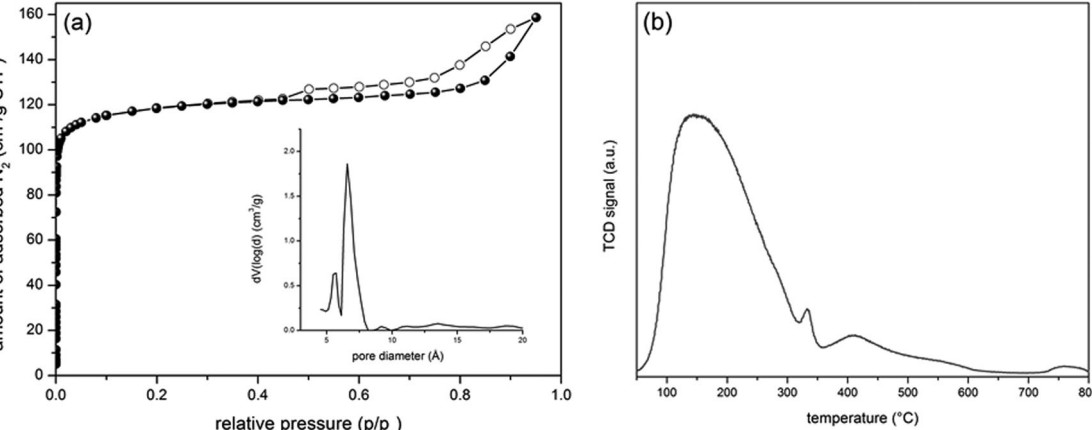

**Fig. 2 Porosity properties of bioNICS-1. a** $N_2$ isotherm measured at 77 K. Adsorption points—full symbols, desorption points—empty symbols. Inset shows pore size distribution obtained by NLDFT method. **b** Temperature-programmed desorption of ammonia profile.

impact of pH. For that purpose, water and saline solutions were adapted to pH values of 3.5, 7.4 and 9.0; (b) the presence of ionic species that are most commonly present in body fluids. For that purpose, saline (0.9% NaCl) and phosphate buffer solution (PBS), in addition to the demineralized water, were used; and (c) the influence of environment dynamics on the framework. Therefore, the stability tests were performed in steady state and under stirring. Structural stability of material was studied with PXRD and BET-specific surface measurements of recovered products in comparison to the untreated material. The rate of released Zn(II) cations within the supernatant was also measured by atomic absorption spectrometry (AAS) analysis in order to better correlate with the degradation, assuming that the detected concentration of released Zn(II) is directly linked with the rate of framework degradation.

In all cases, the majority of the material remains undissolved in water, saline and PBS solutions after 24 h, with its structural integrity remaining generally unchanged after treatment. However, diffraction peak broadening and decrease of their heights occur due to the decrease in crystallite size. The additional appearance of Zn-oxalate phase, which is formed as a consequence of partial degradation of ascorbic to oxalic acid, can be observed as well (Fig. 3a and Supplementary Fig. 12).

(a) pH of the media does not have a marginal effect on the material's degree of solubility or degradation. The pH values of the supernatant settle at around 6 after 24 h, regardless of the initial pH value, type of media and dynamics. This is however confirmed to be normal, since ascorbic acid being released into the supernatant acts as an efficient buffer in wide range of pH values[59]. (b) Evidently, the porosity is preserved to a large extent (up to 68 %) for the products exposed to demineralized water (Supplementary Figs. 14, 16 and Supplementary Table 4). The decrease of the surface area is most probably affected by the presence of Zn-oxalate phase within the bulk material. The presence of additional ionic species affects the pore accessibility even further. That is, by inducing the solubility of the parent MOF due to the increase of the ionic strength, as in the case of saline solution, or by provoking progressive degradation of the MOF framework due to the anion replacement with the ligand, which can take place in phosphate buffer solution[60,61]. As expected, the framework degrades to a larger extent in saline solutions, which is consistent with the trend of BET surface area values and 24 h the degradation rate (release of zinc). Modification of micropores during the degradation tests (Supplementary Fig. 15) follows the same trend as a decrease in BET specific surface area however; average pore size is not significantly affected by dissolution media. The amount of released Zn(II),

however, remains under 10% in all cases (Fig. 3b and Supplementary Fig. 13). Zinc release in PBS solution at dynamic conditions is comparable to the one where the material was exposed to water. On the other side, bioNICS-1 show profoundly higher insolubility (below 1%) in PBS when leaving the mixture undisturbed. STEM-EDXS elemental mapping revealed that Zn phosphate-based shell with a thickness of ~10 nm is formed in such conditions (Fig. 3c) which apparently decelerates subsequent dissolution of the Zn-ascorbate framework. (c) This recrystallization process is apparently more pronounced when the material is subjected to dynamic as opposed to static conditions. Nevertheless, relatively high stability of bioNICS-1, even in acidic or basic starting conditions, is unusual for Zn-based MOF systems and can be attributed to the sum of specific structural features such as high connectivity of the pentadentate ascorbic acid molecule to Zn(II) centres, the high dimensionality of inorganic building units and the rigidity of the framework[62]. The most pronounced effect of dynamics is observed when stirring is applied in PBS solution where ~5% of Zn(II) is released from the framework (Fig. 3b and Supplementary Fig. 13) and recrystallization to oxalate is most notable (Fig. 3a), resulting in a decrease of microporosity to about 28 % of the parent value (Supplementary Figs. 14, 16 and Supplementary Table 4). These results however are reassuring, because they imply that the framework can be readily dissolved in conditions that are (out of tested ones), most closely related to the body's environment. Therefore, future encapsulated drugs are going to be released and constituents of bioMOF further metabolized.

Additional information of the degradation process was investigated by the release of organic (ligand) constituents from bioNICS-1 framework by liquid NMR (Supplementary Fig. 17). The released ascorbic acid in aqueous media (water or saline) in all cases additionally degrades into threonic acid representing a common metabolite of a vitamin C[63]. However, $^{13}C$ NMR spectra of the bioNICS-1 degradation in PBS solutions indicate the presence of two additional unknown C4 molecular species, which most probably occur as a result of threonic acid esterification with phosphate anions.

**Kinetics and mechanism of structure degradation.** In a further attempt to investigate the degradation mechanism of bioNICS-1 under different aqueous conditions, the release kinetics was studied (Fig. 4a). For demineralized water, saline and phosphate buffer solutions the pH was adjusted to 7.4. Zn(II) concentrations were measured by AAS from liquid solutions sampled after

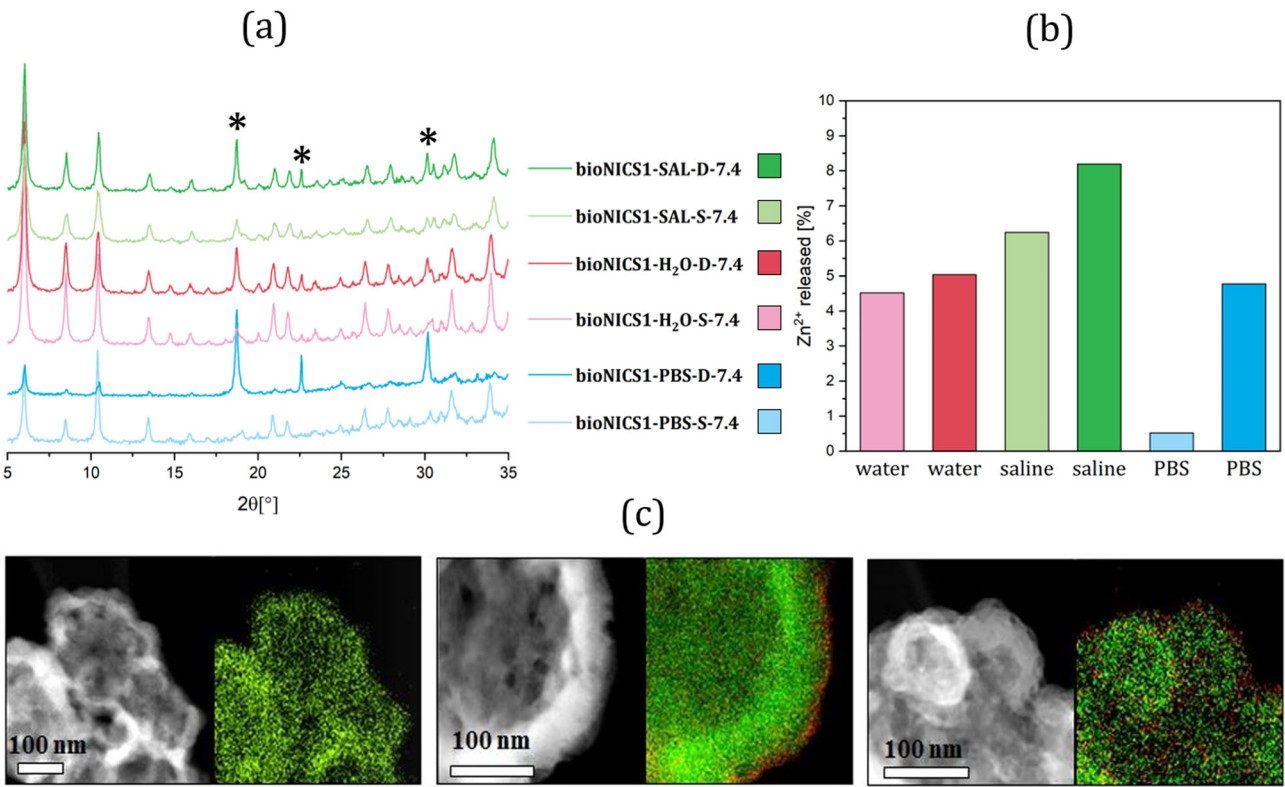

**Fig. 3 bioNICS-1 material stability in different aqueous conditions. a** XRD patterns of the selected products treated under different aqueous conditions. Name indicates treatment conditions: bioNICS1-x-y-7,4 (x—sample immersed in demineralized water ($H_2O$), saline solution (SAL) or phosphate buffer solution (PBS); y—static (S) or dynamic conditions (D); 7,4—pH value of solutions. Diffraction peaks corresponding to Zn-oxalate are indicated by asterisks; **b** Released % of Zn(II) cations from the bioNICS-1 framework within the supernatant calculated on the basis of the Zn(II) concentration measurements by ICP-OAS. **c** STEM-ADF micrographs and EDXS elemental mapping of the parent bioNICS-1 (left), bioNICS-1-PBS-S-7.4 (middle) and bioNICS-1-PBS-D-7.4 (right). The distribution of Zn and P are represented in green and red colour respectively.

specified times of stirring, ranging from 5 min to 7 days. The release of Zn(II) shows similar trends for all used media, which cannot be described with any specific kinetic model established for drug release[64,65]. The Zn(II) cation release is entirely governed by the framework dissolution rate, rather than diffusion through the inert matrix, as presumed by the afore mentioned models. In the case of bioNICS-1, the release profile shows two stages. Initial dissolution of the bioNICS-1 framework within the first 6 h appears as a slight burst release of Zn(II) cations followed by a relatively slow dissolution process in a controllable manner, which follows a concentration-independent zero-order rate profile. The framework dissolution is most likely inhibited by the formation of insoluble Zn(II)-oxo—based species due to the ligand exchange in the shell domains of the crystallite, which significantly slows down the subsequent dissolution of the framework (Fig. 4d). The dissolution rate is, as expected, the fastest in saline solutions and slowest in PBS solutions, releasing 18 and 5% of $Zn^{2+}$ from the bioMOF framework, respectively. The mechanism was confirmed in the case of bioNICS-1 degradation in PBS solution, representing the most biologically relevant medium. The formation of phosphate-based nanodomains on the bioNICS-1 crystallites has already been proven by TEM elemental mapping as described earlier in the text. On the other hand, the XRD patterns of the recovered solids from PBS revealed the gradual formation of additional Zn-oxalate phase, as a result of partial degradation of ascorbic acid to oxalic acid (Fig. 4c). Both phases cannot be complementarily detected using diffraction and microscopic methods. Zn-phosphate domains are of amorphous nature and therefore not detectable by XRD, on the other hand, Zn-oxalate is not reliably distinguishable from Zn-ascorbate

phase by TEM elemental mapping due to the similar zinc content in both investigated phases. The formation of both types of insoluble domains nicely coincides with the release kinetic profile. Initial dissolution of bioNICS-1 framework is reflected in slight burst release of Zn(II) cations within the first 6 h of treatment. Afterwards, the release process stabilizes for the remaining time to PBS exposure in linear progression, due to the formation of phosphate- and oxalate-based phases on the shell domains of the crystallites. In addition, the relative crystallinity of the bioNICS-1 framework deteriorates simultaneously with the recrystallization process of bioNICS-1 to Zn-oxalate (Fig. 4b, and Supplementary Fig. 18). It can be concluded that the release of Zn(II) cations is subjected to a controllable manner for most of the treatment time and apparently becomes governed by the solubility of Zn-oxalate rather than bioNICS-1 itself with increasing time of exposure in the solution. The proposed mechanism can be easily projected to the bioNICS-1 degradation in demineralized water and saline solution as well. The Zn(II) release rates in such conditions are faster as in the case of treatment in PBS solution where the presence of phosphate-based shell domains additionally inhibits the dissolution process.

**Drug loading.** As the intended application of the material is drug delivery and to prove that small-molecule drugs can be encapsulated and subsequently released from the microporous framework, bioNICS-1 was loaded with urea which is a model drug for topical application and cosmetic agent[9,66,67]. Namely, the size of the pores between 5.5. and 6.5 Å, hydrophilic nature of the framework with accessible acid sites as confirmed by ammonia TPD

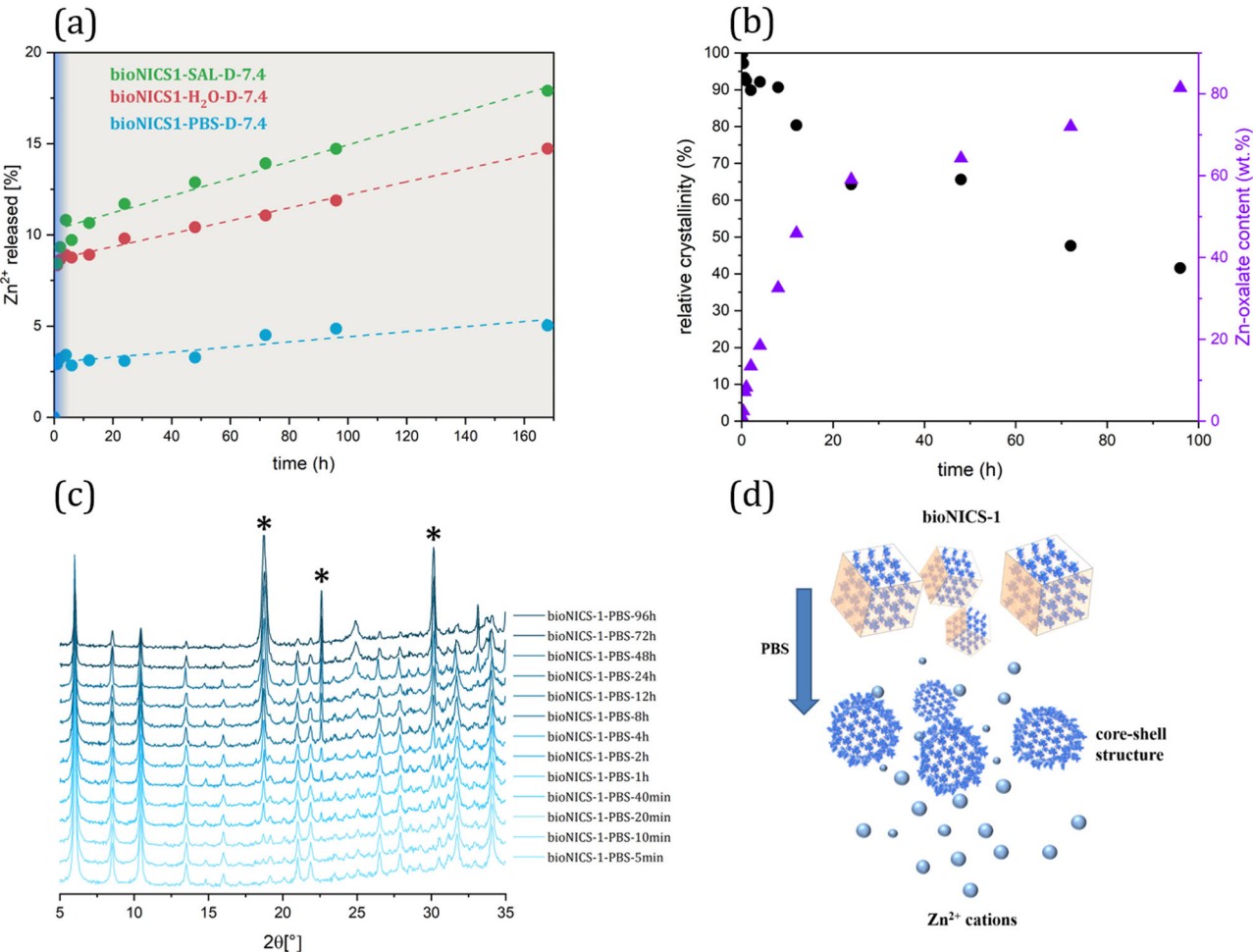

**Fig. 4 Degradation of bioNICS-1. a** Zn(II) release kinetics of the selected bioNICS-1 products treated in demineralized water (H$_2$O), saline solution (SAL) or phosphate buffer solution (PBS) at adjusted pH values of 7.4. Blue area shows the time range where Zn(II) cations are released by burst effect whereas grey area represents the release in near-linear regime indicated by the dashed lines. **b** Relative crystallinities of the bioNICS-1 structures (black dots) and Zn-oxalate weight content (purple triangles) of the solids recovered from PBS solution after stirring at specified times.; **c** XRD patterns of the solids recovered from PBS solution after stirring for specified times. Peaks corresponding to Zn-oxalate are indicated by asterisks; **d** Schematic representation of the proposed mechanism of bioNICS-1 structure degradation with core-shell formation.

and suitable structural stability in water media, makes the bioNICS-1 suitable host for urea loading. Other—small—therapeutic molecules such as hydroxycarbamide, dimethyl fumarate, phosphonomethanoic acid and penicillamine[68] could also be loaded into the pores. Drug loading was done via a simple impregnation method where activated material was dispersed in an ethanol solution of urea for 48 h. The TG/DTG curve of the urea-loaded and dried material (bioNICS-1@urea) show a weight loss of 11.5% in the temperature region between 100 and 220 °C, which is not observed in the TG curve of the pristine bioNICS-1 (Fig. 5a and Supplementary Fig. 20). The efficient encapsulation of urea drug within the micropores of bioNICS-1 is also indicated by the significant decrease of BET specific surface area from 553 to 150 m$^2$/g for the pristine and bioNICS-1@urea materials respectively as deduced from the N$_2$ sorption isotherms (Fig. 5b). The absence of the urea-corresponded peaks in the XRD pattern of loaded material additionally confirms that the drug is confined within the micropores rather than recrystallized separately or on the surface of the material (Supplementary Fig. 19).

The capability of drug release was demonstrated by the immersion of bioNICS-1@urea in demineralized water for 12 h. The urea drug was identified in the supernatant by liquid $^1$H NMR analysis (inset in Fig. 5a and Supplementary Fig. 21).

Quantitative evaluation of the NMR spectra showed that 9–10 wt.% of the loaded material corresponds to the released urea. With other words, 80–90% of the loaded drug is released in an aqueous medium within 12 h, which is triggered by the partial degradation of bioNICS-1 framework as described in details above.

## Conclusion

A new MOF containing bio-compatible Zn(II) cations and ascorbic acid as the main constituents were developed using facile solvothermal synthesis with the use of non-toxic solvent ethanol. The first reported case of Zn-ascorbate MOF (bioNICS-1) is excelled by several unique structural features—diverse Zn(II) coordination geometry forming an infinite three-dimensional inorganic building unit bridged through ascorbate ligands, resulting in three-dimensional ascorbate-based framework structure with permanent microporosity and accessible acid metal sites. BioNICS-1 was thoroughly examined for structural stability and solubility in different aqueous conditions using water, saline and PBS solutions adjusted at pH values from 3.5 to 9 under stirring or static conditions. Framework rigidity and a high degree of Zn(II) binding via all five hydroxyl-type oxygen atoms

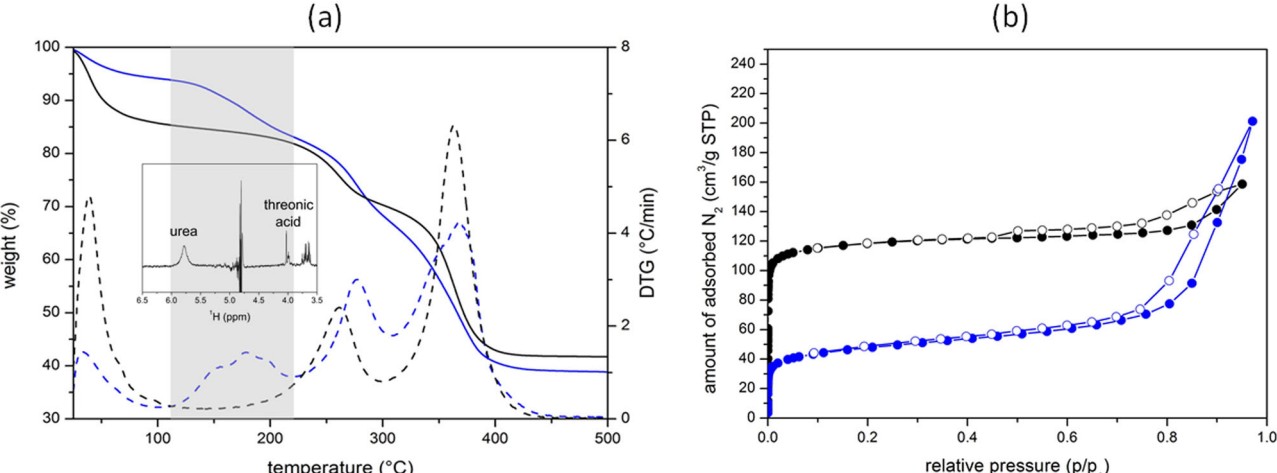

**Fig. 5 Drug loading and release. a** TG/DTG curves of bioNICS-1@urea (blue) and initial bioNICS-1 (black) materials. Grey area represents the temperature range of the urea removal from the materials. Inset shows $^1H$ NMR spectrum of dissolved bioNICS-1@urea in demineralized water indicating the characteristic peak of urea; **b** $N_2$ isotherms of bioNICS-1@urea (blue) and initial bioNICS-1 (black) materials measured at 77 K. Adsorption points—full symbols, desorption points—empty symbols.

provides relatively high structure stability and so enable controllable release of bioactive components. The highest solubility rate is achieved in saline solutions, where ~9% of the framework's Zn(II) cations were released and about 40% of its initial porosity was preserved after 24 h of stirring. On the other hand, phosphate ions in PBS solution significantly inhibit the framework dissolution due to the formation of less soluble oxalate and phosphate-based domains on the surface of crystallites as evidenced by XRD, TEM and NMR analysis. BioNICS-1 solubility is almost pH-independent due to the strong buffering effect of the released ascorbic acid settling the pH of the media at an approximate value of 6. Kinetic studies showed that regardless of the media type, the framework dissolution undergoes two regimes. A slight burst effect is observed within the first 6 h, followed by a slow dissolution described in zero-order kinetics, releasing 5 and 18% of $Zn^{2+}$ from bioNICS-1 in PBS and saline solution respectively, within 7 days. The proposed dissolution of the framework is suppressed by the formation of Zn-oxalate shell domains on the surface of the bioNICS-1 crystallites. The accessibility of the micropores was confirmed by successful impregnation with model drug—urea and its efficient release triggered by the partial framework degradation in aqueous media. Overall, the slow dissolution and recrystallization process of bioNICS-1 with a burst effect, which could no dubitably be well managed with suitable shaping, makes the material an exceptional and highly promising candidate for a controlled small drug delivery system provided by its framework microporosity.

## Methods

**Synthetic approaches**. BioNICS-1 was prepared using EtOH as a solvent with Zn/ASC ratio of 2:1 using conventional heating. Typically 0.752 g of zinc(II) acetate dehydrate (Sigma Aldrich) and 0.3 g of L-ascorbic acid (Sigma Aldrich) was added to 10 ml of ethanol (Sigma Aldrich). The reaction mixture was heated at 120 °C for 1 day in Teflon-lined Stainless-steel autoclaves. Larger independent crystals for the structural analysis were prepared with the addition of acetic acid (Alfa Aesar). The optimum EtOH/acetic acid volume ratio was 10:1 (1 ml of acetic acid in 10 ml of ethanol).

**Characterization methods**. X-ray powder diffraction data of the samples were collected on a PANalytical X'Pert PRO high-resolution diffractometer with CuKa radiation ($\lambda = 1.5406$ Å) in the range from 5 to 60° (2θ) with the step of 0.034° per 100 s using fully opened 100 channel X'Celerator detector. For the purposes of Rietveld refinement of the crystal structure model, the XRD powder data was collected on the same equipment using transmission mode in the range from 5 to

90° 2θ with the step of 0.016°/300 s. Prior to the measurement, the bioNICS-1 sample was degassed at 150 °C overnight and sealed in glass capillary.

Morphological properties and size estimation of the samples was observed by scanning electron microscopy measurements (SEM) on Zeiss Supra™ 3VP field-emission gun microscope. Elemental analysis was performed by energy dispersive X-ray analysis with an INCA Energy system attached to the above described microscope and by Perkin Elmer 2400 Series II CHNS analyser.

The structure of bioNICS-1 was solved from 3DED data, collected using a JEOL JEM-2100 LaB6 transmission electron microscope. Details of the structure analysis are provided in the Supplementary Methods section.

The thermal analysis (TG/DTG) was performed on a Q5000 IR thermogravimeter (TA Instruments, Inc.). The measurements were carried out in air flow of 10 ml/min, by heating samples from 25 to 700 °C at the rate of 10 °C min$^{-1}$. Temperature-programmed X-ray powder diffraction pattern of samples was recorded also on the PANalytical X'Pert PRO diffractometer additionally equipped with a high-temperature sample cell, from room temperature to 500 °C in steps of 50 °C in static air. $N_2$ sorption isotherms measurements were performed on Quantachrome AUTOSORB iQ3. The specific surface areas were determined by Brunauer–Emmett–Teller (BET) method based on the $N_2$ sorption isotherms measured at 77 K in p/p$_o$ relative pressure range between $4 \times 10^{-2}$ and $6 \times 10^{-3}$ selected according to Roquerol plots. Before the measurement, samples were activated under vacuum at 150 °C for 15 h. The temperature-programmed desorption (TPD) experiments were performed using the Micromeritics AutoChem II 2920 apparatus. The sample (120 mg) was positioned inside a U-shaped quartz reactor and pre-treated in a flow Ar at 150 °C for 180 min. After pre-treatment, the latter was cooled to 50 °C and saturated with 10% $NH_3$ in helium for 30 min. Weakly adsorbed $NH_3$ was removed in a flow of pure He for 60 min. The samples were heated to 800 °C at 20 °C/min and $NH_3$ desorption was monitored by a mass spectrometer (Pfeiffer Vacuum Thermostar) following the characteristic $m/z$ fragments. Pulses of 0.5 mL 10% $NH_3$ in He (Messer) were used as an external standard for $NH_3$-TPD signal calibration.

STEM-EDXS elemental mapping was performed on a probe Cs-corrected Jeol ARM 200CF Scanning Transmission Electron Microscope equipped with a Centurio Energy Dispersive X-ray Spectroscopy (EDXS) system with a 100 mm$^2$ SDD detector was used for imaging and acquisition of elemental mappings.

**The aqueous stability test**. 250 mg of activated bioNICS-1 was suspended in 25 mL of purified water and saline solution, pH was regulated with HCl and NaOH to reach the values of 3.5; 7.4 and 9.0. The test was done in parallels; one of them was left undisturbed, the other exposed to dynamic conditions on a circular plane mixer for 24 h. Afterwards the samples were left to settle down before separately collecting the supernatant and the sediment for further analysis. The filtered sediment was left to dry on air over night and its identity was checked and confirmed with PXRD and $N_2$ isotherms analysis. The same procedure was repeated with phosphate buffer solution at pH 7.4.

Supernatants were collected and analysed for the presence of organic compounds by NMR spectroscopy. 10% $D_2O$ was added to 0.5 ml of each sample. The samples were transferred to 5 mm NMR tubes and $^1H$ NMR spectra were recorded on a Bruker Avance Neo 600 MHz NMR spectrometer with a QCI cryo probe at 25 °C using excitation sculpting to suppress the water signal. In addition, $^{13}C$, $^1H$-$^{13}C$ HSQC and $^1H$-$^{13}C$ HMBC NMR spectra were recorded on the sample of supernatant after treatment in phosphate buffer solution under dynamic

conditions to obtain more detail about the nature of compounds in the sample. Chemical shifts were referenced externally to the chemical shift of NaTMSP ($\delta_H$ 0 ppm, $\delta_C$ 0 ppm).

**Zn(II) release kinetics.** The test was a kinetic modification of the previously described procedure presented by Howarth et al. Briefly, 250 mg of activated bioNICS-1 was suspended in 25 mL of 3 different media (purified water, saline and phosphate buffer solution), which represented about 5 mg of Zn/1 mL media. Solutions had pH of 7.4 which was regulated with HCl and/or NaOH. Samples of release medium (5 mL) were collected for analysis at set time intervals (1, 2, 4, 6, 12, 24, 48, 72, 96 and 168 h) using a syringe with a 0.80 μm filter attached, and supplemented with the same volume of fresh media. This way the volume of the release medium stayed constant throughout the test, and *sink* conditions were maintained. The concentration of Zn(II) cations within the investigated solutions were determined by PerkinElmer AAnalyst 200 flame atomic absorption spectrometer (AAS).

The concentration of the released Zn(II) was calculated using Eq. 1, where $C_{t(corr.)}$ is the corrected concentration at the time t, $C_t$ is the apparent concentration at the time t, v is the volume of the sample taken and V is the total volume of the dissolution medium.

$$C_t(corr.) = C_t + \frac{v}{V} \sum_0^{t-1} C_t \qquad (1)$$

**Framework degradation kinetics in PBS.** 85 mg of activated bioNICS-1 was suspended in 8,5 mL of PBS pH 7.4 and immediately exposed to dynamic conditions on a circular plane mixer. At a set time point (5 min, 10 min, 20 min, 40 min, 1 h, 2 h, 4 h, 8 h, 12 h, 24 h, 48 h, 72 h, 96 h) individual samples were filtered, dried on air over night and sediments identity checked with PXRD.

The relative crystallinity of the bioNICS-1 material recovered from PBS solution after different times of exposure was estimated by comparison of peak intensities corresponding to 011 reflection occurring at 6.0° 2θ. The weight contribution of Zn-oxalate which is formed during the above mentioned process was calculated from Rietveld quantification analysis using TOPAS Academic V6 software package[69].

**Drug loading and release experiments.** Activated bioNICS-1 (200 mg) was suspended in EtOH solution of urea (25 mg mL$^{-1}$) for 48 h under constant stirring. Impregnation was done in two parallels, one at room temperature and one at 40 °C. Both residues were filtered and washed with EtOH, then dried on air over night before proceeding with further tests. The recovered product was immersed in demineralized water for 12 h. After that the supernatant was qualitatively and quantitatively evaluated by liquid $^1$H NMR.

## Data availability

The data generated during this study are included in the manuscript and the Supplementary Information.

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

## Acknowledgements
We would like to thank Dr. Janvit Teržan from the National Institute of Chemistry, Slovenia for performing the temperature-programmed desorption experiments. Financing from the Slovenian Research Agency program (P1-0021) is acknowledged.

## Author contributions
T.K.T. and M.M. performed the majority of the synthesis and experimental work, N.Z.L. contributed in manuscript writing, E.S.G. and T.W. performed the structure analysis by ED, T.A.J. performed AAS analysis, U.J. covered NMR measurements, G.D. performed STEM-EDS elemental mapping. All authors have given approval to the final version of the manuscript.

## Competing interests
The authors declare no competing interests.
