## [Peer Review File · Communications Chemistry]

Reviewers' comments:

Reviewer #1 (Remarks to the Author):

This word described a vitamin C-based zinc MOF with permanent pores and its degradation in different aqueous conditions. The bioMOFs constructed from pure bioactive linkers are very rare, and few of them are with permanent pores, such as mediMOF-1 synthesized from curcumin and zinc. The limitation of the pores in this MOF for drug delivery could be the small pore size of 5 and 6 angstrom, since most drugs are large molecules. However, as the authors said, permanent porosity is not always a priority. I recommend the publication of this work only after the questions as following well responded.

Questions:

1. The structure of bioNICS-1 is more like a inorganic-organic hybrid. Its inorganic building unit is 3D. Is it possible that only the inorganic framework retain if the linkers removed?
2. The authors should offer some drugs which could be delivered by bioNICS-1, or other potential applications in biomedicine field using permanently porous frameworks like bioNICS-1.
3. The particle size is an important parameter in biomedical applications. The particle size of bioNICS-1 is between 40-120 nm which is very suitable. The question is whether the nanoparticles could disperse in solvents well.
4. The bioMOFs constructed from pure bioactive linkers and with permanent pores are very few. The related reference should be cited.
5. Some other suggestions: ASC should be explained when it firstly appeared; the text size is too small in Figures.

Reviewer #2 (Remarks to the Author):

This article presents an ascorbic acid based bioMOF as a carrier for drug delivery. Although the idea is interesting and experiment results are in line with expectations, the MOF-based biomaterial for drug delivery is not novel, and the current outline needs are re-organized. Therefore, I think it is hard to recommend this article to be published in this journal. It should be processed completely before the submission is considered.

Here are my detailed comments:

1. There are a few grammatical errors and the language is not native enough. Thus, the English in this manuscript needs to be further improved.
2. Many characterization experiments of bioNICS-1-aa were conducted, but they lacked in-depth analysis and discussion between the structure and function. Therefore, the data needs to be rearranged to better integrate them.
3. In line 80, the authors stated that zinc ions could not promote the production of undesirable reactive oxygen or nitrogen species, but it is inconsistent with the current antibacterial mechanism of zinc ions. A further explanation should be provided.
4. The design and color scheme in many Figures were so disharmonious that the detailed information cannot be identified, such as Figure 3b, Figure 4c, Figure 5a, Figure S12, and Figure S16. The authors need to modify them to make it easier for readers to extract critical information from them.
5. Much necessary information is missing throughout the manuscript, such as the absent scale bars in Figure S1b, Figure S1c, and the full name of DLS in line 135. The authors should check the

manuscript carefully to avoid similar mistakes.

Reviewer #3 (Remarks to the Author):

The manuscript presents a Zn(II) bioMOF based on vitamin C with structural design of 3D porosity, which could result in slow degradation and controlled release properties. The results are of some novelty, can potentially be used for drug delivery. Some conclusions could be further strength by additional elucidation or experiments.

1. What is major advantage of using ascorbic acid over nicotinic acid? Other than no one has used L-ascorbic acid before?
2. Figure 1c-d is quite difficult to understand. P6 Line 150-153, it this hypothesis? What is its contribution to permanent porosity? How would it relate to the SEM characterization of pores?
3. Does Vc-Zn MOF with no structural design also degrades? Does the structure of bioNICS-1 improve the degradation and solubility? Will the permanent pores change during degradation?

Reviewers' comments:

Reviewer #1 (Remarks to the Author):

This word described a vitamin C-based zinc MOF with permanent pores and its degradation in different aqueous conditions. The bioMOFs constructed from pure bioactive linkers are very rare, and few of them are with permanent pores, such as mediMOF-1 synthesized from curcumin and zinc. The limitation of the pores in this MOF for drug delivery could be the small pore size of 5 and 6 angstrom, since most drugs are large molecules. However, as the authors said, permanent porosity is not always a priority. I recommend the publication of this work only after the questions as following well responded.

Questions:

1. The structure of bioNICS-1 is more like an inorganic-organic hybrid. Its inorganic building unit is 3D. Is it possible that only the inorganic framework retain if the linkers removed?

Authors response:

The attempt to synthesize the inorganic material isostructural to bioNICS-1 in the absence of ascorbic acid was done as well. However, this attempt was unsuccessful. This is now addressed in the manuscript (Page 6 Line 167). Ascorbic acid, even though not being the part of topological description, is necessary for the formation of this unique framework.

The framework structure does not withstand ligand removal as it is clearly shown by temperature-programmed XRD (see Supplementary information Figures S10 and S11, Page 15). Elimination of ascorbic acid results in formation of dense ZnO.

2. The authors should offer some drugs which could be delivered by bioNICS-1, or other potential applications in biomedicine field using permanently porous frameworks like bioNICS-1.

Authors response:

The drug that can be delivered is governed by its kinetic diameter and is apparently limited by the size of the drug carrier pores. BioNICS-1 exhibit microporosity with the size of the channels between 5.5 and 6.5 Å as described in the manuscript. In this case the drug delivery is limited to 'small' molecules such as therapeutic gasotransmitters (NO, H₂S, CO). Besides that, there are several non-gaseous bioactive active substances with the sufficient molecular dimensions that are widely used in medical applications, such as As₂O₃ and hydroxyurea used in cancer therapy, fomepizole (4-methylpyrazole) used in methanol poisoning treatment, 4-aminopyridine used for multiple sclerosis treatment, etc. All mentioned drugs require controlled (hindered) release within the organism and therefore bioNICS-1 could be their potential carrier after proper structure modification overcoming the agglomeration challenges and thorough toxicological tests. The potential use of other small-molecule drug is now addressed in the manuscript (Page 8 Line 198, Page 14 Line 339).

At the current level of bioNICS-1 research described herein, the research was more focused in the direction of topical applications; therefore the urea was used as a model drug and to proof the concept of pore accessibility for such small drug. Namely, urea acts as a skin moisturizer and is used for various skin treatments.

3. The particle size is an important parameter in biomedical applications. The particle size of bioNICS-1 is between 40-120 nm which is very suitable. The question is whether the nanoparticles could disperse in solvents well.

Authors response:

Dispersion in water media was tested and the particles can be re-dispersed. This is additionally addressed in the manuscript (Page 6 Line 142). At the current stage of the bioNICS-1 investigation, this is however not considered as an important issue. Moreover, with the proper surface modification (many different paths are well established in the literature), the change in the surface charge or creating the steric hindrance can greatly inhibit the agglomeration. This is a subject of a current investigation and goes beyond the scope of the submitted manuscript.

4. The bioMOFs constructed from pure bioactive linkers and with permanent pores are very few. The related reference should be cited.

Authors response:

Thank you for the comment. As already emphasized in the manuscript, even though the permanent porosity adds an important value to the bioMOF application, this structural property is still a rarity in the field. With the additional references describing porous bioMOFs, which are now included in the manuscript (Page 4 Line 94), we believe that representative systems defined as bioMOFs, are now covered.

5. Some other suggestions: ASC should be explained when it firstly appeared; the text size is too small in Figures.

Authors response:

Thank you for your suggestions. This is now corrected.

Reviewer #2 (Remarks to the Author):

This article presents an ascorbic acid based bioMOF as a carrier for drug delivery. Although the idea is interesting and experiment results are in line with expectations, the MOF-based biomaterial for drug delivery is not novel, and the current outline needs are re-organized. Therefore, I think it is hard to recommend this article to be published in this journal. It should be processed completely before the submission is considered.

Here are my detailed comments:

1. There are a few grammatical errors and the language is not native enough. Thus, the English in this manuscript needs to be further improved.

Authors response:

Grammar was revised and errors were corrected.

2. Many characterization experiments of bioNICS-1-aa were conducted, but they lacked in-depth analysis and discussion between the structure and function. Therefore, the data needs to be rearranged to better integrate them.

Authors response:

Thank you for the suggestions. BioNICS-1-aa was synthesized using acetic acid as modulator in order to design larger and separated crystallites which were suitable for structure analysis (solving the structure) using 3DED technique. Structure analysis proved that acetic acid does not interfere with the framework, neither it is encapsulated within the pores, since no additional electron density was indicated in the Patterson map. Rietveld refinement which was based on the powder XRD pattern performed on a nanosized bioNICS-1 sample (Supporting information, Figure S4) show that the calculated pattern from the obtained structure model and the experimental data are agreeably matching ($R_{wp} = 0.07$). Comparison of basic characterization techniques (XRD, TG, N_2 isothermal measurements) between bioNICS-1 and bioNICS-1-aa prove that both products reside the same structure with matching characteristics. The structural analogy enables us to use only the product synthesized conventionally (bioNICS-1) containing nanosized (more relevant for application) particles for further detailed studies of degradation in aqueous media. BioNICS-1-aa exhibit lower specific surface area as initial sample, suggesting the limited accessibility for hosting molecules when the larger crystal are grown. This again points out the relevance and advantages of nanoparticle design when materials are considered for bioapplications. Additional structural analysis of bioNICS-1-aa is now added in the Supplementary information for comparison (Figure S5).

3. In line 80, the authors stated that zinc ions could not promote the production of undesirable reactive oxygen or nitrogen species, but it is inconsistent with the current antibacterial mechanism of zinc ions. A further explanation should be provided.

Authors response:

We thank the reviewer for the comment. The statement indeed requires further explanation. Zinc toxicity and its effect on ROS and RNS generation is now more clearly explained and supported with additional literature. Antimicrobial activity mechanism which is explained through the generation of ROS or RNS is in fact studied for ZnO nanoparticles and not for Zn^{2+} cations themselves. Zn^{2+} is in contrast to ZnO nanoparticles considered as an antioxidant mineral protecting against oxidative

stress, so the generation of reactive oxidative species is at least within the frames of RDI values, inhibited rather than induced. This is now explained more clearly in the manuscript (Page 3 Line 80).

4. The design and colour scheme in many Figures were so disharmonious that the detailed information cannot be identified, such as Figure 3b, Figure 4c, Figure 5a, Figure S12, and Figure S16. The authors need to modify them to make it easier for readers to extract critical information from them.

Authors response:

Thank you for your suggestions. The colour schemes are now more consistent.

5. Much necessary information is missing throughout the manuscript, such as the absent scale bars in Figure S1b, Figure S1c, and the full name of DLS in line 135. The authors should check the manuscript carefully to avoid similar mistakes.

Authors response:

The required corrections were done.

Reviewer #3 (Remarks to the Author):

The manuscript presents a Zn(II) bioMOF based on vitamin C with structural design of 3D porosity, which could result in slow degradation and controlled release properties. The results are of some novelty, can potentially be used for drug delivery. Some conclusions could be further strength by additional elucidation or experiments.

1. What is major advantage of using ascorbic acid over nicotinic acid? Other than no one has used L-ascorbic acid before?

Authors response:

From the crystal structure engineering perspective, there are several points where ascorbic acid can be considered advantageous over the nicotinic acid: (1) ascorbic acid offers higher number of linkage to the metal centres through four hydroxyl groups and one keto-group, thus providing greater topological variety in respect to the nicotinic acid where coordination is possible only via pyridine and carboxylate groups; (2) Slightly bulkier molecule of ascorbic acid if compared to the niacin could also provide larger spacing between metal nodes and therefore greater possibility to design frameworks with permanent porosity; (3) Buffering property of ascorbic acid, which is already discussed in the manuscript, stabilizes the pH in the solvothermal process which usually have a crucial role in the thermodynamics of the crystallization. 'Self-controlled' system that is provided by buffering effect, can have an important impact on product reproducibility and/or yields and opens more manoeuvring space for structure modification (i.e. addition of modulators, surfactants,...) without significantly affecting the thermodynamic equilibrium within the reaction system.

From the physiological perspective it is really hard to discuss about the advantages of the individual molecule, since each has its specific and vital role in the organism. This point is now emphasized in the manuscript (Page 4 Line 108). The detailed study of pharmacological differences of the two vitamins are however beyond the scope of this paper.

2. Figure 1c-d is quite difficult to understand. P6 Line 150-153, it this hypothesis? What is its contribution to permanent porosity? How would it relate to the SEM characterization of pores?

Authors response:

Figure 1 caption is now revised and hopefully written more clearly.

As described in the manuscript, the coordination diversity of Zn(II) cation which bioNICS-1 structure exhibit is caused by high linking ability of ascorbic acid - four hydroxyl groups and one carbonyl group. High degree of dentation, which is very rare among the ligands, forms a diverse coordination environment of the central cation within the same structure framework which is also very rarely observed. Relation between multi-dentation ability and coordination diversity is considered to be a fact rather than hypothesis.

Inorganic building units (shown in Figure 1 c in the manuscript) are stacked in the rigid framework in such way that they form micropores. Such arrangement is possible only in the presence of ascorbic acid. Therefore, the linking of inorganic part with vitamin C is mandatory for permanent porosity. SEM method cannot be used for evaluation of these pores, due to too the low spatial resolution of the technique. On the other hand, HR-TEM could be more appropriate for such analysis, however MOFs are generally facing the problems of framework sensitivity to the high-energy focused electron beams. Finally, pore size evaluation extracted from N₂ sorption data is considered to be sufficiently reliable.

3. Does Vc-Zn MOF with no structural design also degrades? Does the structure of bioNICS-1 improve the degradation and solubility? Will the permanent pores change during degradation?

Authors response:

Molecular compound of Zn and vitamin C with no structural design into 3-dimensional structures (MOFs) most probably degrades in uncontrollable manner. The intention of structural design is to hinder degradation into their constituents and make the release more controllable. Therefore the solubility is expected to be lower than in the case of un-designed molecular compounds.

Size and shape of the pores are not significantly affected by dissolution media used in solubility studies. This is now included in Supplementary information (Figure S15). Degradation process merely decreases the specific surface area of the material due to the formation of insoluble nonporous shell domains. The pore size distribution becomes more broad and the micropore volume decreases only

when the material is exposed to the PBS solution. This is somehow expected, due to the additional recrystallization of ascorbate phase into phosphate-based domains as proven by NMR.

REVIEWERS' COMMENTS:

Reviewer #2 (Remarks to the Author):

Authors have addressed my concerns about the manuscript, and the manuscript is suitable for publication in this journal.

Reviewer #3 (Remarks to the Author):

Comments have been addressed.